Antimalarial target vulnerability of the putative Plasmodium falciparum methionine synthase

Leela Nirut 1
Prommana Parichat 2
Kamchonwongpaisan Sumalee 2
Taechalertpaisarn Tana 1
Shaw Philip J. philip@biotec.or.th 2
1 Department of Microbiology, Faculty of Science, Mahidol University , Bangkok , Bangkok , Thailand
2 National Center for Genetic Engineering and Biotechnology (BIOTEC), National Science and Technology Development Agency , Pathum Thani , Thailand
Nunes-da-Fonseca Rodrigo
Electronic publication date: 2024 Jan 15
Publication date: 2024
Volume: 12
Electronic Location ID: e16595
Received 2023 Jun 30; Accepted 2023 Nov 14
Copyright: ©2024 Leela et al.
Copyright year: 2024
Copyright holder: Leela et al.
License: This is an open access article distributed under the terms of the Creative Commons Attribution License, which permits unrestricted use, distribution, reproduction and adaptation in any medium and for any purpose provided that it is properly attributed. For attribution, the original author(s), title, publication source (PeerJ) and either DOI or URL of the article must be cited.
License URL: https://creativecommons.org/licenses/by/4.0/

Keywords: Plasmodium falciparum, Methionine synthase, Malaria, GlmS riboswitch, GlmS ribozyme, Target vulnerability

Funding: National Science and Technology Development Agency, Thailand #P18-50116 #P2052577 This work was supported by the National Science and Technology Development Agency, Thailand (Research Chair grant #P18-50116 awarded to Sumalee Kamchonwongpaisan and team, and grant #P2052577 awarded to Sumalee Kamchonwongpaisan). The funders had no role in study design, data collection and analysis, decision to publish, or preparation of the manuscript.

==============================
Background

Plasmodium falciparum possesses a cobalamin-dependent methionine synthase (MS). MS is putatively encoded by the PF3D7_1233700 gene, which is orthologous and syntenic in Plasmodium. However, its vulnerability as an antimalarial target has not been assessed.

Methods

We edited the PF3D7_1233700 and PF3D7_0417200 (dihydrofolate reductase-thymidylate synthase, DHFR-TS) genes and obtained transgenic P. falciparum parasites expressing epitope-tagged target proteins under the control of the glmS ribozyme. Conditional loss-of-function mutants were obtained by treating transgenic parasites with glucosamine.

Results

DHFR-TS, but not MS mutants showed a significant proliferation defect over 96 h, suggesting that P. falciparum MS is not a vulnerable antimalarial target.

Introduction

Malaria is a devastating parasitic disease. Between 2019 and 2021, an estimated additional 13.4 million cases were attributed to malaria control disruptions, chiefly the COVID-19 pandemic (World Health Organization, 2022). Infections with Plasmodium falciparum are responsible for most malaria cases, which are treated with artemisinin combination therapy (ACT). Artemisinin-resistant P. falciparum parasites are widespread throughout Southeast Asia, and partially resistant parasites have evolved independently in parts of Africa (World Health Organization, 2022), prompting the need for antimalarials against novel targets. Given that the current standard of care for malaria is a three-day ACT regimen, any new antimalarial must be similarly fast-acting. Fast-acting antimalarials inhibit the functions of essential, vulnerable targets (Forte et al., 2021). More than half of the protein-coding genes in P. falciparum are annotated as essential based on the criterion that disruption of the protein-coding region by transposon insertion is not tolerated (Zhang et al., 2018). However, the absence of transposon insertion is not definitive annotation of essentiality because of local variation in transposon insertion efficiency. One possible example of this scenario is the PF3D7_1311700 (cyt c-2) gene, which lacks transposon insertions but was shown to be dispensable by targeted knockout (Espino-Sanchez et al., 2023).

Identifying vulnerable targets is challenging because some essential genes are non-vulnerable, including targets for which specific inhibitors with antimalarial activity are available, such as deoxyhypusine synthase (Aroonsri et al., 2019), Niemann-Pick Type C1-Related protein (Istvan et al., 2019), and plasmepsin V (Sleebs et al., 2014; Polino et al., 2020). Partial loss-of-function (LOF) mutants of these genes have latent proliferation defects, and in the case of plasmepsin V, a defect was observed after 96 h only in LOF mutants with greater than 90% knockdown of the wild-type level of expression (Polino et al., 2020). Antimalarial discovery efforts might be better focused on directly identifying vulnerable targets with alternative assays rather than proving essentiality, which may require laborious monitoring of proliferation in LOF mutants with varying degrees of knockdown and/or conditional knockout mutants over extended periods for non-vulnerable targets. We propose to define vulnerable targets for the purpose of assay development as genes for which a partial LOF mutant (with significant knockdown of about 50 to 90% reduction of the wild-type expression level) has an acute proliferation defect observable at 96 h or sooner.

In this study, we developed a target vulnerability assay for LOF mutants created with the glmS ribozyme tool (Prommana et al., 2013). To apply the tool, the gene of interest must be modified by DNA transfection. Proof of concept for the tool was previously demonstrated for the dihydrofolate reductase-thymidylate synthase (DHFR-TS) gene modified by single-crossover integration of transfected circular DNA (Prommana et al., 2013; Aroonsri et al., 2016). This transfection method has been superseded by the more efficient CRISPR-Cas9 gene editing system (Ghorbal et al., 2014). We edited the DHFR-TS gene to assess whether the glmS ribozyme was functional in the context of an edited gene, particularly one with the 3′ coding region replaced with artificial recodonized sequence as a consequence of gene editing. In addition, the LOF mutant obtained with an edited DHFR-TS gene was used to validate the target vulnerability assay. DHFR-TS is a known vulnerable target that is inhibited by antifolate drugs such as pyrimethamine and P218 (Yuthavong et al., 2012), and LOF mutants of DHFR-TS show significant proliferation defects at 72 h or earlier (Prommana et al., 2013; Aroonsri et al., 2019).

To search for new antimalarial targets, we propose testing LOF mutants of unexplored genes in target vulnerability assay. Methionine metabolic pathways contain several unexplored antimalarial targets. Methionine is an essential amino acid in P. falciparum that must be obtained from salvage because proliferation is markedly reduced in culture media lacking methionine (Divo et al., 1985; Marreiros et al., 2023). P. falciparum salvages methionine via the new permeation pathway and a neutral amino acid transporter (Cobbold, Martin & Kirk, 2011). In addition to protein synthesis, methionine is used as a cofactor to produce the essential metabolite S-adenosyl-l-methionine (SAM) by the SAM synthase (SAMS) enzyme. The methyl group from SAM is transferred to various acceptors by methyltransferases to form S-adenosyl-l-homocysteine (SAH). SAH is hydrolyzed to adenosine and l-homocysteine (Hcy) by the highly conserved enzyme S-adenosyl-l-homocysteine hydrolase (Tanaka et al., 2004). P. falciparum lacks key enzymes in the reverse transsulfuration pathway for the conversion of Hcy to cysteine. Consequently, Hcy accumulates and is effluxed from the parasite during intra-erythrocytic growth (Beri et al., 2017). However, excess Hcy is deleterious and can trigger gametocytogenesis in P. falciparum (Beri et al., 2017). In addition to efflux for the control of Hcy, Hcy can be converted to methionine by the action of the methionine synthase enzyme (MS, 5-methyl tetrahydrofolate homocysteine methyltransferase, EC.2.1.1.13). MS uses 5-methyltetrahydrofolate (5-mTHF) as a cofactor to generate tetrahydrofolate (THF) as a by-product (Banerjee & Matthews, 1990). P. falciparum can obtain 5-mTHF from salvage or by synthesis from 5,10 methylenetetrahydrofolate via a methylenetetrahydrofolate reductase enzyme (Asawamahasakda & Yuthavong, 1993).

Cobalamin-dependent MS enzymatic activity was reported previously in P. falciparum intra-erythrocytic stage protein extract. Nitrous oxide inhibits the activity of this enzyme and parasite proliferation, suggesting that parasite cobalamin-dependent MS may be a potential antimalarial target (Krungkrai, Webster & Yuthavong, 1989). However, the gene encoding the parasite MS enzyme has not yet been identified in the P. falciparum 3D7 genome (Müller & Hyde, 2013). We identified a candidate P. falciparum MS gene and created a LOF mutant for assessing the vulnerability of this target.

Materials & Methods

Ethical approval

Blood for parasite culture was obtained by a protocol approved by the Ethics Committee, National Science and Technology Development Agency (NSTDA), Thailand, approval document #0021/2560. Written consent was obtained from all volunteers.

Bioinformatic analyses

The InterPro database (Paysan-Lafosse et al., 2023) was searched using the InterPro entry IPR003726 (homocysteine-binding domain) via the InterPro web interface (https://www.ebi.ac.uk/interpro/). Protein sequences were obtained from UniProt (The UniProt Consortium et al., 2023) of the cobalamin-dependent methionine synthase (MS) enzymes from human (MTR, Q99707-1) and Escherichia coli K12 (metH, P13009), together with Plasmodium spp. candidate MS from orthologous group 1324at5820 encoded by P. falciparum PF3D7_1233700 (Pf, Q8I585), P. knowlesi PKH_145080 (Pk, A0A384KWI2), P. malariae PmUG01_14067900 (Pm, A0A1A8X239), P. ovale wallikeri PowCR01_140053700 (Pow, A0A1C3L5P3), P. ovale curtisi PocGH01_14059300 (Poc, A0A1D3UAH7), and P. vivax PVX_100640 (Pv, A0A1G4H5F2) genes. Sequences were aligned using the T-Coffee tool with default settings in the Expresso web interface (Armougom et al., 2006).

The X-ray structure determined to 1.90 Å resolution of the Hcy/5-mTHF binding fragment of Thermotoga maritima cobalamin-dependent MS co-complexed with Hcy and 5-mTHF (PDB: 1q8j; Evans et al., 2004) was used as a query for searching proteins with homologous structures in the Plasmodium falciparum 3D7 proteome. The PDB accession number was provided as a query source to the Foldseek web tool (Van Kempen et al., 2023; https://search.foldseek.com/search). Target search was restricted to the AlphaFold/Proteome v4 P. falciparum 3D7 database of 5,187 ab initio predicted protein structures using the 3Di/AA mode under default settings.

Construction of transfection vectors

Cas9 vectors were constructed by cloning oligonucleotides containing the guide RNA (gRNA) sequence (Table S1) into the pDC2-Cas9-hDHFRyFCU plasmid (Knuepfer et al., 2017) digested with BbsI (New England Biolabs (NEB) Ipswich, MA, USA). To construct the repair vectors, we first made a mother plasmid (p3HA_glmS; Dataset S1) with glmS and HA elements to regulate and monitor the target protein, respectively. Repair vectors (Dataset S2 & Dataset S3) were constructed by simultaneous Gibson assembly cloning (NEB) of three synthetic fragments for each target (recodonized partial open reading frame, 5′ and 3′ HR) into the p3HA_glmS plasmid, linearized by digestion with KpnI (NEB). Recodonized coding region fragments were obtained as gBlocks synthetic DNA (IDT, Singapore). 5′ and 3′ HR fragments were obtained by PCR using PrimeSTAR® GXL DNA Polymerase (Takara Bio Inc. Shiga, Japan) and P. falciparum 3D7 genomic DNA template.

Gene editing by DNA transfection

P. falciparum 3D7 reference (NCBI txid: 36329) wild-type parasite was cultured in vitro as previously described (Aroonsri et al., 2016; Aroonsri et al., 2019). Cas9 (20 µg) and PstI-linearized repair (50 µg) vectors were co-transfected into late schizont parasites by AMAXA nucleofection, as previously described (Knuepfer et al., 2017). WR99210 (2 nM, a gift from Prof. Tirayut Vilaivan) was applied 48 h post-transfection to select transfected parasites. WR99210-resistant parasites emerged within 30 days post-transfection, and gene editing events were detected by PCR with integration-specific primers (Table S1). Transfected parasites with detectable gene edits were treated for 7 days with 1 µM 5-fluorocytosine (Sigma-Aldrich, Merck KGaA, Darmstadt, Germany) and 2.5 µg/mL blasticidin S HCl (Gibco™, Thermo Fischer Scientific, Waltham, MA, USA) to remove Cas9 plasmid retained as episome and eliminate wild-type parasites. Clonal lines of gene-edited parasites were obtained by limiting dilution in 96-well microtiter plates.

Western blotting

Clonal lines of transgenic parasites DHFR-TS_glmS and MS_glmS with edited PF3D7_0417200 and PF3D7_1233700 genes, respectively were synchronized by sorbitol treatment and cultured for 24 h in the presence or absence of 5 mM glucosamine (GlcN, Sigma-Aldrich, Burlington, MA, USA). Parasites liberated from host cells by saponin treatment were lysed in RIPA Lysis and Extraction Buffer (Thermo Fisher Scientific, Waltham, MA, USA), sonicated for 15 s, and centrifuged at 14,000 g for 5 min. The supernatant was harvested, and the total protein concentration was determined by BCA protein assay (Pierce, Thermo Fisher Scientific). A sample of protein extract (50, 50, and 2.5 µg of total protein from 3D7 wild-type, MS_glmS, and DHFR-TS_glmS transgenic parasites, respectively) was separated in each lane of a NuPAGE 4–12% Bis-Tris protein gel in MOPS running buffer (Invitrogen, Waltham, MA, USA) using an XCell Surelock Electrophoresis cell (Invitrogen). Proteins were transferred onto a 0.45 µm PVDF transfer membrane (Thermo Fisher Scientific) by electroblotting using an XCell blot module (Invitrogen). The membrane was stained using LI-COR REVERT™ 700 total protein stain (LI-COR Biosciences, Lincoln, NE, USA). The membrane was blocked in Odyssey® blocking buffer (LI-COR Biosciences) overnight and probed with primary antibody (Anti-HA-Tag Rabbit Monoclonal antibody # SAB5600116, diluted 1: 50,000; Sigma-Aldrich) for 1 h. After washing three times, the membrane was incubated with IRDye 800CW goat anti-rabbit IgG (LI-COR Biosciences, diluted 1: 20,000) for 1 h. Blots were analyzed using the Odyssey® CLx Infrared Imaging System (LI-COR Biosciences). Total protein and target protein band (HA-tagged MS = 72.7 kDa and HA-tagged DHFR-TS = 75.5 kDa) intensities were determined using Image Studio v5.2 software (LI-COR Biosciences). Lane normalization factors were determined from the total protein signal (700 nm channel) in each lane. The target protein band intensities (800 nm channel) were adjusted using the lane normalization factors. GlcN-treated lane-factor adjusted intensities were normalized to the corresponding signals of untreated parasites from the same experiment (100%), which were used for quantitative analysis. The % target protein (GlcN treated relative to untreated control) signals were analyzed using two-tailed one-sample Welch’s t-tests in R Statistical Software (v4.3.0; R Core Team, 2023), comparing the sample means with a null hypothesis mean of 100%. The P-values from t-statistics were adjusted using the Holm-Bonferroni post-hoc method in R. The mean DHFR-TS and MS % target protein signals were compared using two-tailed two-sample Welch’s t-tests in R with the null hypothesis of no means difference.

Parasite proliferation (target vulnerability) assay

P. falciparum parasites were cultured for 96 h in 96-well microtiter plates at different GlcN concentrations. Parasite proliferation was assessed using SYBR Green I fluorescence as previously described (Aroonsri et al., 2016). The background-subtracted SYBR Green I signals from GlcN-treated parasites were normalized to the average background-subtracted signal from control parasites from the same synchronized culture without GlcN (100%) and were taken as response values for analysis. Data from at least three independent experiments for each parasite line were analysed using the drc package version 3.0-1 (Ritz & Streibig, 2005) in R v4.3.0 with the four-parameter log–logistic model. The top and bottom values were fixed at 100 and 0, respectively. The slope and 50% response (EC50) values were fitted separately for each parasite line. EC50 values for each gene-edited transgenic line were compared with that of the 3D7 wild-type strain using the EDcomp function in the drc R package. The P-values from t-statistics reported by EDcomp were adjusted using the Holm-Bonferroni post-hoc method in R.

Results

We hypothesized that P. falciparum possesses an MS-encoding gene. We searched for P. falciparum 3D7 proteins with an Hcy-binding domain in the InterPro database (Paysan-Lafosse et al., 2023), since all MS enzymes possess an N-terminal Hcy-binding domain (Matthews, Sheppard & Goulding, 1998). The PF3D7_1233700 gene product is the only P. falciparum 3D7 protein with an InterPro-annotated Hcy-binding domain. PF3D7_1233700 is annotated in OrthoDB (Kuznetsov et al., 2023) as a member of the syntenic Plasmodium orthologous group 1324at5820 (Hcy-binding domain). PF3D7_1233700 and orthologous (single-copy) proteins from other human-infective Plasmodium spp. showed low (≈20%) identity with human and Escherichia coli cobalamin-dependent MS (Figs. S1 & S2). We searched for candidate MS using a three-dimensional structural superposition-based approach (Foldseek), which is more sensitive for identifying homologous proteins (Van Kempen et al., 2023). Full-length cobalamin-dependent MS comprises four modules (N-terminal Hcy-binding, 5-mTHF-binding, cobalamin-binding, and C-terminal adenosylmethionine-binding/reactivation; Matthews, Sheppard & Goulding, 1998). The X-ray structures of cobalamin-dependent MS protein fragments from different species have been determined, although the structure of the N-terminal fragment containing Hcy- and 5-mTHF substrate binding domains is available only for Thermotoga maritima (Evans et al., 2004). We selected the structure of the T. maritima cobalamin-dependent MS fragment co-complexed with Hcy and 5-mTHF as a query for Foldseek. The top-ranked Foldseek hit to P. falciparum 3D7 proteins was PF3D7_1233700. Lower-ranked hits had well-defined annotations unrelated to methionine metabolism, suggesting incidental structural similarity of protein folds with functions unrelated to MS (Table S2). Notwithstanding the possibility of even more diverged proteins not detectable by sequence- or structure-based homology, PF3D7_1233700 is putatively assigned as P. falciparum MS. However, definitive annotation requires direct functional data, e.g., biochemical assay of the purified PF3D7_1233700 protein for MS activity.

We edited the PF3D7_1233700 and PF3D7_0412700 (DHFR-TS) genes, placing them under the control of the glmS ribozyme (Prommana et al., 2013). The edited DHFR-TS and MS genes were confirmed by PCR genotypic assays in clonal lines of transgenic parasites (Figs. 1 & 2). One clonal line of each edited gene (designated as DHFR-TS_glmS and MS_glmS, respectively) was selected for phenotypic analysis. The expressions of target proteins in transgenic parasites were assessed by western blotting of synchronized transgenic parasites cultured for 24 h in the presence or absence of GlcN. Protein species of sizes expected for modified MS (72.7 kDa) and DHFR-TS (75.5 kDa) were detected in transgenic parasites (Fig. S3). GlcN treatment caused significant reductions in MS and DHFR-TS protein expression (DHFR-TS mean = 40%, P-adjusted = 0.004; MS mean = 51%, P-adjusted = 0.03; Fig. 3A). There was no significant difference in the mean % target protein (GlcN treated relative to untreated control) level between DHFR-TS and MS (P = 0.34).

Figure 1 PF3D7_0417200 (DHFR-TS) gene editing.

(A) Schematic of PF3D7_0417200 gene editing (drawn to scale). The location of the guide RNA target for mediating double-strand DNA break is indicated by the lightning bolt symbol. Repair vector (Dataset S2) elements are indicated by the colored boxes, including homologous regions in blue, recodonized protein coding region (Rec.) in magenta, triple hemagglutinin epitope tag (HA) in cyan, glmS ribozyme (glmS) in red, blasticidin S deaminase selectable marker gene (BSD) in orange, and Plasmodium transcriptional regulatory elements in teal. The locations and sizes of amplicons expected from PCR using primers DHFR_37 F (P1), DHFRTS_1558R (P2), and glmS_3R (P3) are indicated by black arrows. The structures of the PF3D7_0417200 gene before editing in parental reference strain 3D7 wild-type (3D7 WT) and after editing in transgenic (DHFR-TS_glmS) parasites are shown. (B) PCR products from genotypic assay separated in 0.8% agarose gel. Three clonal lines of transgenic parasites were isolated; DHFR-TS_glmS clone #1 was selected for phenotypic analysis. Lane M: 1kb+ DNA ladder (Invitrogen, sizes indicated on the left).

Figure 2 PF3D7_1233700 (MS) gene editing.

(A) Schematic of PF3D7_1233700 gene editing (drawn to scale). The location of the guide RNA target for mediating double-strand DNA break is indicated by the lightning bolt symbol. Repair vector (Dataset S3) elements are indicated by the colored boxes, including homologous regions in blue, recodonized protein coding region (Rec.) in magenta, triple hemagglutinin epitope tag (HA) in cyan, glmS ribozyme (glmS) in red, blasticidin S deaminase selectable marker gene (BSD) in orange, and Plasmodium transcriptional regulatory elements in teal. The locations and sizes of amplicons expected from PCR using primers MS_5IntF (P1), MS_HR2_rev (P2), glmS_3R (P3), and MS_5recodonR (P4) are indicated by black arrows. The structures of the PF3D7_1233700 gene before editing in parental reference strain 3D7 wild-type (3D7 WT) and after editing in transgenic (MS_glmS) parasites are shown. (B) PCR products from genotypic assay separated in 0.8% agarose gel. Two clonal lines of transgenic parasites were isolated; MS_glmS parasite clone #1 was selected for phenotypic analysis. Lane M: 1kb+ DNA ladder (Invitrogen, sizes indicated on the left).

Figure 3 Phenotypic analysis of gene-edited parasites.

In vitro cultures were established for Plasmodium falciparum reference 3D7 parental strain and transgenic parasite strains MS_glmS and DHFR-TS_glmS with edited PF3D7_1233700 (MS) and PF3D7_0417200 (DHFR-TS) genes, respectively. (A) Knockdown of target proteins in transgenic parasites. MS and DHFR-TS % target protein signals were obtained by western blotting (Fig. S3). Boxplots show the data from triplicate experiments. (B) Target vulnerability assay. Parasites were cultured for 96 h at different glucosamine (GlcN) concentrations. The left panel shows all data and model fits (curves). The right panel shows EC_50 values for each transgenic line compared with that of the 3D7 parental strain. The points show the estimated EC_50 ratio (3D7: transgenic parasite) and error bars represent S.E.M. The dashed line indicates the line of no effect. Estimated EC_50 ratios:- 3D7: DHFR-TS_glmS = 11.25, adjusted P = 7.9E-6; 3D7: MS_glmS = 0.89, adjusted P = 0.84.

Next, we assessed the consequences of target protein knockdown in transgenic parasites with respect to proliferation in target vulnerability assay. In previous studies of the acute effect of glmS-ribozyme mediated target knockdown on parasite proliferation, treatment was performed for up to 72 h in which GlcN has a minor inhibitory effect on wild-type strains (Prommana et al., 2013). By extending the GlcN treatment to 96 h, greater than 50% inhibition of the 3D7 wild-type strain was observed at the highest concentrations such that we could determine the EC50 (8.0 mM; 95% confidence intervals 5.9 to 10.0 mM). We posited that knockdown of vulnerable target gene expression by the action of the glmS ribozyme enhances the proliferation defect caused by GlcN treatment itself over 96 h manifested as a significantly lower EC50 compared with 3D7 wild-type. The EC50 of the DHFR-TS_glmS parasite, but not that of the MS_glmS parasite, was significantly different from that of the 3D7 wild-type strain (Fig. 3B). Hence, reduction in DHFR-TS, but not MS expression affected parasite sensitivity to GlcN. Based on these results, DHFR-TS is defined as a vulnerable antimalarial target as expected. In contrast, MS is a non-vulnerable target.

Discussion

PF3D7_1233700 was identified as the only candidate gene encoding MS from a bioinformatic search of the P. falciparum 3D7 genome, suggesting that the parasite possesses a single MS gene. However, P. falciparum MS is not a vulnerable antimalarial target, in contrast to the vulnerability of the downstream SAMS enzyme in the P. falciparum methionine pathway (Musabyimana et al., 2022). Salvage is the major source of methionine substrate for P. falciparum SAMS because reducing exogenous methionine leads to a concomitant decrease of SAM (Harris et al., 2023). Hence, methionine synthesized by MS is of minor importance for intra-erythrocytic proliferation under standard in vitro culture conditions. However, methionine synthesis may be more important for P. falciparum proliferation in natural infections, since the methionine concentration in human serum (Barić et al., 2004) is approximately two to seven times lower than that of parasite culture medium.

The other roles of P. falciparum MS besides methionine synthesis should be considered to explain the conservation of MS in Plasmodium and its tentative annotation of essentiality in P. falciparum based on the absence of transposon insertions in the encoding gene (Zhang et al., 2018). It should be noted that definitive proof of essentiality requires demonstration of a proliferation defect from more complete knockdown using a different tool (e.g., the TetR-DOZI system with 5′ and 3′ aptamers installed at the target gene (Polino et al., 2020)) or from conditional gene knockout mutagenesis. Essential genes are expressed through the life cycle, and P. falciparum MS is detectably expressed by data-independent proteomics throughout the intraerythrocytic stages at a level approximately 40–1000-fold lower than that of DHFR-TS (Siddiqui et al., 2022). Moreover, single-cell RNA sequencing data indicate that P. falciparum MS is expressed during mosquito stages (Real et al., 2021), and the P. vivax MS ortholog PVP01_1451800 is expressed in liver stages (Mancio-Silva et al., 2022).

Although malaria parasites possess a mechanism for the efflux of Hcy (Beri et al., 2017), lack of MS function may lead to increased levels of Hcy and redox stress, which may be important for sporogonic development in the mosquito vector when Plasmodium is more dependent on glutathione (Vega-Rodríguez et al., 2009) and α-lipoic acid (Biddau et al., 2021) to mitigate oxidative stress. The conversion of 5-mTHF to THF by MS may be important for recycling folate required for other enzymatic reactions, particularly during the developmental stages with the greatest folate demand. Plasmodium can salvage folates and the folate precursor para-aminobenzoic acid (p ABA), but the levels of folates are too low, or are not in a form capable of being efficiently transported to support intra-erythrocytic development in the absence of de novo synthesis (Salcedo-Sora & Ward, 2013).

P. berghei lacking the p ABA synthetic enzyme aminodeoxychorismate synthase cannot develop in p ABA-deficient medium during intraerythrocytic stages; however, the growth of these mutant parasites is unaffected in p ABA-deficient medium during liver stages (Matz et al., 2019). Despite an inefficient folate transport system for the uptake of 5-mTHF (Salcedo-Sora & Ward, 2013), elevated levels of 5-mTHF in the liver may drive its accumulation and conversion to THF by Plasmodium MS, such that the parasite is less reliant on de novo folate synthesis during this stage of the life cycle. A knockout mutant of the P. berghei orthologous MS gene (PBANKA_1448300) shows no growth defect during intraerythrocytic stages (Bushell et al., 2017), but growth of the mutant is reduced during the transition from the sporozoite (through the liver) to the blood stage (Stanway et al., 2019). Although these data suggest a non-essential role of Plasmodium MS, dispensability in P. falciparum cannot be extrapolated from knockout data in P. berghei because of species-specific differences in Plasmodium host cell tropism. P. berghei preferentially invades reticulocytes, whereas P. falciparum invades mature erythrocytes. The reticulocyte milieu has a greater metabolic complexity than that of erythrocytes, which can support the growth of P. berghei parasites with knockouts of genes functioning in the intermediary carbon metabolic pathway, pyrimidine metabolism, and glutathione biosynthesis that are essential in P. falciparum (Srivastava et al., 2015).

Conclusions

The finding that MS is a non-vulnerable antimalarial target raises the question of what other enzymes in the Plasmodium parasite methionine pathway and other pathways related to folate metabolism (Müller & Hyde, 2013) are also non-vulnerable targets. This could be tested by target vulnerability assay of LOF mutants for other genes annotated as essential. Although the role of MS as an antimalarial target is deprioritized, it would be interesting to test whether the roles of P. falciparum MS in Hcy metabolism and folate recycling are more important in mosquito and liver stages.

Supplemental Information

Supplemental Information 1 Sequence alignment of methionine synthases

Protein sequences were obtained from UniProt of the cobalamin-dependent methionine synthase (MS) enzymes from human (MTR, Q99707-1) and Escherichia coli K12 (metH, P13009), together with Plasmodium candidate MS from orthologous group 1324at5820 encoded by P. falciparum PF3D7_1233700 (Pf, Q8I585), P. knowlesi PKH_145080 (Pk, A0A384KWI2), P. malariae PmUG01_14067900 (Pm, A0A1A8X239), P. ovale wallikeri PowCR01_140053700 (Pow, A0A1C3L5P3), P. ovale curtisi PocGH01_14059300 (Poc, A0A1D3UAH7), and P. vivax PVX_100640 (Pv, A0A1G4H5F2) genes. Sequences were aligned using the T-Coffee tool with default options in the Expresso web interface (Armougom et al., 2006).

Click here for additional data file.

Supplemental Information 2 Identity matrix of methionine synthases

Percent identity matrix of the protein alignment (Fig. S1) was created using the bio3d R package version 2.4-4. Cells are shaded according to the percent identity as indicated by the scale bar on the right.

Click here for additional data file.

Supplemental Information 3 Western blotting of target proteins

Clonal lines of transgenic parasites DHFR-TS_glmS and MS_glmS with edited PF3D7_0417200 (DHFR-TS) and PF3D7_1233700 (MS) genes, respectively were cultured for 24 h in the presence or absence of 5 mM glucosamine (GlcN). A sample of parasite protein extract (50, 50, and 2.5 µg of total protein from 3D7 wild-type, MS_glmS and DHFR-TS_glmS transgenic parasites, respectively) was separated in each lane of a 4–12% NuPAGE Bis-Tris protein gel in MOPS running buffer (Invitrogen). Upper panel shows total protein staining with REVERT (700 channel). Lower panel shows target protein signal of the same membrane detected with anti-HA antibody (800 channel). The images are uncropped and unedited. Total protein and target protein band (HA-tagged MS = 72.7 kDa and HA-tagged DHFR-TS = 75.5 kDa) intensities were determined using Image Studio v5.2 (LI-COR Biosciences). Migrations of DHFR-TS and MS target protein bands are indicated on the right. Lane designations:- Lane 1: iBright™ Prestained Protein Ladder (Invitrogen, Thermo Fischer Scientific) Lane 2: 3D7 wild-type Lane 3: MS_glmS parasite (-) GlcN (replicate #1) Lane 4: MS_glmS parasite (+) GlcN (replicate #1) Lane 5: MS_glmS parasite (-) GlcN (replicate #2) Lane 6: MS_glmS parasite (+) GlcN (replicate #2) Lane 7: MS_glmS parasite (+) GlcN (replicate #3) Lane 8: MS_glmS parasite (-) GlcN (replicate #3) Lane 9: DHFR-TS_glmS parasite (-) GlcN (replicate #1) Lane 10: DHFR-TS_glmS parasite (+) GlcN (replicate #1) Lane 11: DHFR-TS_glmS parasite (-) GlcN (replicate #2) Lane 12: DHFR-TS_glmS parasite (+) GlcN (replicate #2) Lane 13: DHFR-TS_glmS parasite (-) GlcN (replicate #3) Lane 14: DHFR-TS_glmS parasite (+) GlcN (replicate #3) Lane 15: 3D7 wild-type

Click here for additional data file.

Supplemental Information 4 Oligonucleotide primer sequences

Sequences of primers used for cloning and PCR genotyping.

Click here for additional data file.

Supplemental Information 5 Foldseek results

The X-ray structure of the homocysteine and 5-methyltetrahydrofolate binding fragment of Thermotoga maritima cobalamin-dependent methionine synthase co-complexed with Hcy and 5-mTHF (Evans et al., 2004) was used as a query for searching proteins with homologous structures in the Plasmodium falciparum 3D7 proteome. The PDB accession number (1q8j) was inputted as a query source to the Foldseek web tool (Van Kempen et al., 2023). Target search was restricted to the AlphaFold/Proteome v4 P. falciparum 3D7 database of 5,187 ab initio predicted protein structures using the 3Di/AA mode. Targets are ranked by Foldseek scores.

Click here for additional data file.

Supplemental Information 6 Nucleotide sequence of p3HA_glmS mother plasmid used for construction of repair vectors

Color key:- Yellow: unique 6 bp restriction sites; Cyan: 3x haemagglutinin (HA) epitopes; Gray: RNA spacer; Red: glmS ribozyme (Bacillus subtilis); Dark blue: Plasmodium berghei DHFR-TS terminator; Blue: Plasmodium falciparum histidine-rich protein 2 terminator; Green: Blasticidin-S-deaminase open reading frame; Brown: Plasmodium falciparum calmodulin promoter.

Click here for additional data file.

Supplemental Information 7 Repair vector sequence for PF3D7_0417200 (DHFR-TS) gene editing

Color key:- Yellow: unique 6 bp restriction sites; Cyan: 3x haemagglutinin (HA) epitopes; Gray: RNA spacer; Red: glmS riboswitch (Bacillus subtilis); Dark blue: Plasmodium berghei DHFR-TS terminator; Blue: Plasmodium falciparum histidine-rich protein 2 terminator; Green: Blasticidin-S-deaminase open reading frame; Brown: Plasmodium falciparum calmodulin promoter; Magenta: DHFR-TS 3′homology region; Purple: DHFR-TS 5′homology region; Black: recodonized DHFR-TS.

Click here for additional data file.

Supplemental Information 8 Repair vector sequence for PF3D7_1233700 (MS) gene editing

Color key:- Yellow: unique 6 bp restriction sites; Cyan: 3x haemagglutinin (HA) epitopes; Gray: RNA spacer; Red: glmS ribozyme (Bacillus subtilis); Dark blue: Plasmodium berghei DHFR-TS terminator; Blue: Plasmodium falciparum histidine-rich protein 2 terminator; Green: Blasticidin-S-deaminase open reading frame; Brown: Plasmodium falciparum calmodulin promoter; Magenta: MS 3′homology region; Purple: MS 5′homology region; Black: recodonized MS.

Click here for additional data file.

Supplemental Information 9 Western blot raw data

Raw data from western blotting shown in Fig. 3A.

Click here for additional data file.

Supplemental Information 10 Proliferation assay raw data

Raw data from proliferation assay shown in Fig. 3B.

Click here for additional data file.

We thank Prof. Tirayut Vilaivan (Chulalongkorn University, Bangkok, Thailand) for the gift of WR99210 and Dr. Ellen Knuepfer (Crick Institute, London, UK) for transfection plasmids and protocols.

Additional Information and Declarations

Competing Interests

Author Contributions

Human Ethics

Data Availability

The authors declare there are no competing interests.

Nirut Leela performed the experiments, analyzed the data, prepared figures and/or tables, authored or reviewed drafts of the article, and approved the final draft.

Parichat Prommana performed the experiments, analyzed the data, prepared figures and/or tables, and approved the final draft.

Sumalee Kamchonwongpaisan analyzed the data, authored or reviewed drafts of the article, and approved the final draft.

Tana Taechalertpaisarn conceived and designed the experiments, analyzed the data, authored or reviewed drafts of the article, and approved the final draft.

Philip J. Shaw conceived and designed the experiments, analyzed the data, prepared figures and/or tables, authored or reviewed drafts of the article, and approved the final draft.

The following information was supplied relating to ethical approvals (i.e., approving body and any reference numbers):

Blood for parasite culture was obtained by a protocol approved by the Ethics Committee, National Science and Technology Development Agency (NSTDA), Thailand, approval document #0021/2560. Written consent was obtained from all volunteers.

The following information was supplied regarding data availability:

The images from Western blotting experiments, nucleotide sequences of DNA vectors used in gene editing experiments, and raw data from phenotypic analysis of gene-edited parasites are available in the Supplemental Files.

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
