# Peer review of "Antimalarial target vulnerability of the putative Plasmodium falciparum methionine synthase"

_PeerJ, doi:10.7717/peerj.16595_

## Round 0.1 · original submission · Major Revisions

Dear Dr. Shaw,

Please address carefully the comments of the reviewers, particularly regarding experimental design and controls. Also please address the reason for studying gene PF3D7_0412700 as reviewer 1 pointed out.

Reviewer 1 ·

Basic reporting

In the manuscript the authors showed how MS is not a vulnerable antimalarial target while editing two genes. The hypothesis and question are reasonable but I have a number of points to raise-

1. The authors mentioned two genes that they edited where they gave a reasonable explanation of PF3D7_1233700 gene, but not really about PF3D7_0412700. I do not understand where and how this gene came in the consideration for gene editing. This gene is not present in supplemental table S2 either. The authors need to explain this major point.

2. The authors mention in line 196 that- Notwithstanding the possibility of even more diverged proteins not detectable by sequence- or structure-based homology, PF3D7_1233700 is putatively assigned as
P. falciparum MS. This statement seems to be overstated. The authors need to explain more and give more proofs of why this statement is suitable.

Experimental design

The Experimental design seems fine with a valid research question and hypothesis in my opinion. A few points though-

1. The manuscript lacks to mention the version of the R packages used. Kindly fix that.

2. It will be very helpful if the authors include a workflow of the steps taken in form of a figure.

Validity of the findings

The manuscript delivers a good amount of work but I have some major concerns in addition to aforementioned points-
1. Line 231-
"PF3D7_1233700 was identified as the only candidate gene encoding MS from a bioinformatic search of the P. falciparum 3D7 genome, proving the hypothesis that the parasite possesses a single MS gene."

Overstated. The bioinformatics based searches vary a lot based on parameters. The research may indicate something but saying it is proved is a bit of overstatement.

2. All the figures in manuscript have really lengthy legend. The authors need to take care of that by making plots more elaborative.

3. Fig 1 A. - Labels are confusing over the box. Probably showing with lines/arrows will be better.

4. Fig1 A- Colored boxes that represent MS repair vector need a thorough explanation in the main text.
5. Fig1A- P1-P3 need to be displayed better.
6. Fig1 B- Kindly label lane numbers on the plot instead of feeding too much text in the legend.
7. Fig3A- The authors mention in the legend- "Boxplots show the data from
triplicate experiments. GlcN treatment caused signiûcant reductions in the DHFR-TS (mean
=40%, P-adjusted=0.004) and MS (mean = 51%, P-adjusted=0.03) target signals."
Looking at the plot seems like DHFR-TS is more reduced and MS is relatively higher than DHFR-TS. The authors need to make it clearer and explain about this in the main text more.
8. Kindly prune the legends and feed more texts in methodology instead of figure legends.

Reviewer 2 ·

Basic reporting

This part is sufficient.

Experimental design

This part is sufficient.

Validity of the findings

The conclusion cannot be reached based on the data presented in the manuscript. Please see the attached review for detail.

Annotated reviews are not available for download in order to protect the identity of reviewers who chose to remain anonymous.

Reviewer 3 ·

Basic reporting

The manuscript by Nirut Leela and colleagues performed bioinformatic analysis and identified PF3D7_1233700 as the only gene encoding the likely methionine synthase in the P. falciparum 3D7 genome. Furthermore, they have assessed the targetability of cobalamin-dependent methionine synthase as an antimalarial drug candidate. The authors show evidence that when expression of the methionine synthase is knocked down, there is no growth defect suggesting a non-essential function. However, the experiment design of the current study could be more robust, as outlined below.

Experimental design

1. Given the absence of a growth phenotype of the MS knockdown, it might be valuable to include PCR validation with a primer pair to confirm the absence of WT parasites. This could be a primer pair that works on WT genomic DNA to yield a product but not in the edited locus of clone 1, or to yield an amplicon of two different sizes as shown for the DHFR editing in Figure 2.

2. Western blotting for the confirmation of the protein downregulation is not clear. We may have missed it but there is no legend for supplementary figures to understand the different lanes of the Western blot. If possible, in addition to using total protein as a loading control, detection of another unrelated protein could would also lend weight to the quantitation.
Supplementary figure 3A -please include labelling of the protein ladder.


3. Although no growth defect of the MS line was observed after 96h, it may be that a longer assay could reveal a more subtle defect. Growth assay for longer durations should be performed. Sometimes less protein amount is also enough for parasite survival for few cycles.

Validity of the findings

Identification of PF3D7_1233700 as gene encoding methionine synthase in P. falciparum and analysing its targetability is quite interesting. However the authors may wish to add to the discussion some additional comments regarding further experiments to validate the hypothesis. Specifically, a more precise definition of the term ‘vulnerability’ would be helpful, as they define MS as a non-vulnerable target and DHFR-TS as a vulnerable target. However, based on the experiments shown here, what is shown is a ~50% reduction in MS protein levels does not impact growth, but that is not to say a complete knockout or a more drastic reduction might kill the parasite.

Additional comments

1. The schematics in Figure 1 and 2 can be a bit hard to follow - could more labels be included directly on the figure?
2. Line 199-201 - Absence of piggyBac transposon insertions does not always indicate essentiality, as some non-essential genes are known to be missing insertions by chance.
3. Labelling in Figure 3 could be made clearer – in Figure 3A is the Y-axis a percentage? In Figure 3B right panel, is the X-axis in M?
4. Line 205: The introduction of DHFR-TS comes a bit suddenly and it is not clear why DHFR-TS was chosen for modification along with MS. Perhaps a line explaining the rationale would help the reader.

---

## Round 0.2 · Minor Revisions

Dear Dr. Shaw,

Please address the comments of a single reviewer particularly addressing what would constitute a vulnerable target and the distinction between vulnerability and essentiality should be clarified.

Reviewer 1 ·

Basic reporting

See comments

Experimental design

See comments

Validity of the findings

See comments

Additional comments

The manuscript is revised properly. I appreciate the authors organizing the figures and legends better. I am overall satisfied with the rebuttal submitted by the authors.

Reviewer 3 ·

Basic reporting

No comment

Experimental design

No comment

Validity of the findings

No comment

Additional comments

The authors have satisfactorily responded to the queries and made a number of changes that have improve the manuscript. The one area that could still benefit from more clarity is the concept of a vulnerable or non-vulnerable target, which I agree is a useful distinction when considering drug targets. Although the introduction has been considerably expanded, it still remains a bit unclear what would constitute a vulnerable target – is it one that when knocked down shows a growth defect in the first 96h? Or one that shows a shift in glucosamine EC50? Both are implied in different parts of the paper. And there would have to be a caveat about the level of knockdown. In this instance both proteins are reduced by about 50%, but what if another target was only reduced by 25% and showed no effect. Could that still be considered non-vulnerable if no growth/EC50 effect is observed? Just some additional clarity on the distinction between vulnerability and essentiality (which is less ambiguous) would help.

---

## Round 0.3 · accepted · Accept

Congratulations on the acceptance of your manuscript.